# Source Areas as a Key Factor Contributing to the Recovery Time of Controlled Feral Pigeon (*Columba livia* var. *domestica*) Colonies in Low-Density Urban Locations

**DOI:** 10.3390/ani12091056

**Published:** 2022-04-19

**Authors:** Miguel Ángel Farfán Aguilar, Jesús Duarte, Francisco Díaz-Ruiz

**Affiliations:** 1Departamento de Biología Animal, Facultad de Ciencias, Universidad de Málaga, Campus de Teatinos, 29071 Málaga, Spain; pacodi1480@hotmail.com; 2Instituto IBYDA, Centro de Experimentación Grice-Hutchinson, Loma de San Julián 2, Barriada de San Julián, 29004 Málaga, Spain; 3Ofitecma Marbella S.L., Av. Ramón y Cajal 17, 29601 Marbella, Spain; jddofitecma@gmail.com

**Keywords:** cage-trapping, immigration, recolonization, pest management, urban environments

## Abstract

**Simple Summary:**

Today, governments and administrations strive to minimise issues associated with Feral Pigeon (*Columba livia* var. *domestica*) colonies in urban areas, primarily in large city centres. Scientific evidence has demonstrated that control measures are ineffective in the long term, and colonies recover rapidly. Moreover, very few studies have been conducted in residential zones where colony densities are lower, but where Feral Pigeons generate the same issues. Our primary objective was to evaluate the contributions of the following factors to the recovery time of the Feral Pigeon colonies in residential zones: (1) the distance to the closest area where a separate colony of Feral Pigeons was present, i.e., a source area, and (2) the total or partial removal of the previously existing colony. The distance to the nearest uncontrolled colony of Feral Pigeons was the primary factor that contributed to the recovery time, which significantly increased with increasing distance to the source colonies. Our results highlight the relevance of identifying an effective management unit for the implementation of control programmes to reduce immigration rates and increase long-term effects.

**Abstract:**

Today, governments and administrations strive to minimise issues associated with Feral Pigeon (*Columba livia* var. *domestica*) colonies in urban areas. Scientific evidence has demonstrated that control measures are ineffective in the long term, and colonies recover rapidly. Most scientific research has occurred under high-density circumstances, primarily in large city centres. Moreover, very few studies have been conducted in residential zones or suburban areas where colony densities are lower, but where Feral Pigeons generate the same issues. In this study, we analysed the recovery time of Feral Pigeon colonies in 11 buildings in low-density urban areas where control campaigns were previously conducted to reduce their abundance. Recovery times were highly variable among the buildings (50–3072 days). Distance to the nearest uncontrolled colony of Feral Pigeons, i.e., a source area, was the primary factor that contributed to recovery time, which significantly increased with increasing distance to source colonies. Thus, buildings closest to the Pigeons’ source areas (<500 m) were recolonised more rapidly than were buildings that were >500 m away from source areas. Our findings highlight the relevance of identifying an effective management unit for the implementation of control programmes to reduce immigration rates and increase long-term effects.

## 1. Introduction

The Feral Pigeon (*Columba livia* var. *domestica*) is perfectly adapted to city habitats and is one of the most common bird species in cities across the globe [1]. Factors associated with the success of this species include the rapid incorporation of young into the breeding population (young become sexually active at six months of age) [2], an extended breeding season that covers the entire year [3], high availability of food in urban areas [4,5,6], and low predation rates [7,8,9,10,11]. The global population of Feral Pigeons is estimated to comprise between 165 and 330 million individuals [12]. The success of this species and the high colony densities present in numerous cities [13] have resulted in several important issues. First, Feral Pigeons are carriers of various pathogens that, in many cases, constitute a serious threat to human health. The transmission of pathogens to humans in cities has been reported on several occasions and is a serious risk for immunocompromised persons [14,15,16,17]. Second, Feral Pigeon faeces physically damage buildings and monuments due to the action of bacterial, fungal, plant, and chemical acids present in the droppings [6,18,19,20,21]. Third, relatedly, Pigeon faeces are the primary cause of dirt on facades and internal spaces, as well as unpleasant odours [6,22]. Finally, consequently, governments and administrations must invest great expense to minimise health issues and alleviate infrastructural damage associated with high-density colonies of Pigeons in urban areas [23,24].

To reduce Feral Pigeon colonies to acceptable levels (i.e., abundances for which there are no complaints from neighbours and/or property managers), competent public institutions authorise various control methods such as direct culling, decreasing reproductive success, repellents, audio and visual raptor presence, falconry, reducing the carrying capacity of cities, or live trapping via the use of cage traps [25,26,27,28,29,30,31] Aside from reducing the carrying capacity, which appears to have a lasting effect over time, the remaining control methods are short-lived [30,32]. Moreover, these methods are effective only while they are being applied [2,5,33,34,35], and when a control campaign ends, the colonies recover to initial levels in a short time, which can be as soon as a few days in high abundance areas [34].

In general, control measures are applied in the centre of large cities (>100,000 inhabitants) where Feral Pigeon colonies reach the highest densities. Thus, most scientific studies have been conducted in this setting, and studies are lacking for residential zones or isolated buildings in suburban areas where the density of Feral Pigeon colonies is relatively low [31] but where Feral Pigeons generate the same issues. Similar to the results of studies conducted in higher density areas, Farfán et al. [31] demonstrated that after a certain amount of effort, Pigeon control methods significantly reduced the local abundance of Feral Pigeons in low-density areas. To date, however, the recovery time (defined as the time that elapses between the end of a control programme and the beginning of a new one due to issues generated via an increase in the Feral Pigeon colony) or minimum management area necessary to increase the success of the control campaign have not been studied in this circumstance. In this study, we analysed the recovery time of Feral Pigeon colonies in low-density urban areas where a control campaign was previously conducted to reduce their abundance. Based on current scientific knowledge [32,34,36], we identified two factors as the main modulators of recovery time: (1) the distance to the closest area where separate colonies of Feral Pigeons were present and (2) the total or partial removal of the previously existing colony. Our primary objective was to evaluate the contribution of both factors to the recovery time of the Feral Pigeon colonies. We hypothesised that increased distance to unmanaged colonies (source colonies) would delay the recovery time of managed colonies. In addition, we expected that this distance could be modulated by the home ranges of Feral Pigeons in urban environments [34], where the daily rate of movement of Feral Pigeons is less than 500 m [34,37]. Furthermore, we hypothesised that recovery time would be reduced when a remnant of Pigeons remained in the managed colonies after the end of the control programme, thereby reactivating reproduction. Finally, we aimed to further the discussion of management implications in light of our results.

## 2. Materials and Methods

### 2.1. Study Area

This study was conducted between 2003 and 2010 in three different municipalities (Málaga, Benahavís and Marbella) in the coastal area of the Málaga province in southern Spain (Figure 1). The climate in this region is Mediterranean, with average temperatures of 12 °C in January and 25.5 °C in July and annual precipitation of 534 mm [38,39].

The control of Feral Pigeons was conducted independently in 11 buildings located in seven sites included in urban areas, where hunting is not authorized by current hunting legislation, between the city of Estepona in the west and the city of Málaga in the east. The control campaigns were carried out in each building at different times during the study period (Figure 1 and see Appendix A). In all cases, the buildings were multi-storey buildings (2–4 floors) located in residential zones or isolated areas and had gardens that were also surrounded by large green areas and golf courses, which are the main feeding sites for Feral Pigeons (Appendix A). The density of buildings was extremely low in the study area, and all buildings were relatively far from each other (mean ± SD 20.1 ± 8.1 km) and from the main human population centres such as the cities of Marbella, Fuengirola, Torremolinos, and Málaga (mean ± SD 5.7 ± 3.0 km). Similarly, the population density in the residential zones of the study area was also low (mean ± SD 362.0 ± 267.1 km^2^).

The community of predators in the study area is very small, and the Booted Eagle (*Hieraaetus pennatus*) and the Yellow-legged Gull (*Larus michaellis*) are the only species that occasionally prey on Feral Pigeons.

### 2.2. Feral Pigeon Data

The Feral Pigeon dataset for the study was provided by a private wildlife management company. We then compiled and organised the obtained data to investigate various aspects concerning the recovery time of Feral Pigeon colonies after a control programme. The Feral Pigeons were captured using cage traps and were transferred alive to an industrial dovecote dedicated to Pigeon breeding. The control campaigns were carried out in each building at different seasons, mainly in summer (see Appendix A). To obtain the initial abundance of Feral Pigeons in each building, before starting the control campaign, an observer recorded for a period of 30 min all individuals observed in the building and in its surroundings within a 100 m radius (see Appendix A). For a detailed description of the control programme, see Farfán et al. [31].

The trapping programme was conducted by a private company specialising in wildlife management. The programme followed the recommendations of the European Convention for the Protection of Pet Animals, which has been ratified by the Spanish government, and Law 11/2003 for the Protection of Animals in the autonomous region of Andalusia, which ensures that Feral Pigeons, among other animals, have suitable and sufficient food and water and are maintained in hygienic conditions. All procedures were carried out with appropriate permits provided by the concerned institutions (Ethics Committee of the University of Malaga CEUMA No.: 9-2022-A).

### 2.3. Analysis of Variables

We defined the recovery time (Rt) as the time that elapsed between the end of a control programme and the beginning of a new one due to issues generated via an increase in the Feral Pigeon colony. Based on current scientific knowledge [32,36], our original hypothesis was that the Rt in our study area was primarily modulated by two key factors:

The first of these factors was the distance to a source, i.e., the distance to the nearest uncontrolled colony of Feral Pigeons, which could represent a source area. To calculate the distance (in metres) to the closest Feral Pigeon colony, at the beginning of the control campaign, an observer visually surveyed the surrounding buildings to detect the presence of Feral Pigeons. Once a colony was located and georeferenced with a GPS device, the Google Earth program was used to estimate the distance. If no colony of Pigeons was observed in a building, the distance to the closest known colony was used once it was located by the field technicians.

The second factor related to our hypothesis was the exhaustiveness of the control programme, i.e., the abundance of Feral Pigeons remaining in each building where the control programme was conducted after the control programme ended. In a manner similar to [8,9], a simple method was used to estimate the abundance. Thus, at the end of the control programme, field technicians recorded the number of Feral Pigeons in the building where the control was conducted [31].

The underlying assumption of our hypothesis was that there is a positive relationship between Rt and the distance to a source area and between Rt and the exhaustiveness of the control programme. Thus, the greater the distance to the source and the greater the exhaustiveness of the control, the greater the Rt. To test these relationships, we investigated the Rt in each building year of our study area based on the two key factors mentioned above.

### 2.4. Data Analysis

Control actions were developed on two different occasions for all buildings, and only there were four buildings where controls were developed on at least three occasions (see Appendix A). We considered these repetitions as independent observations in order to develop a simple analysis approach. Generalised Linear Models (GLMs) with a Gaussian distribution of errors and identity link function were employed to test how the Rt was affected by the distance to a source area (i.e., the nearest colony of Pigeons, hereafter referred to as ‘distance’) and the abundance of Pigeons at the end of the control programme (hereafter referred to as ‘Pigeon abundance’). In order to avoid violations of normality and variance homogeneity, the dependent variable (Rt) was transformed using the natural logarithm [40]. We compared all possible combinations of these two independent effects (distance and Pigeon abundance) and their interaction, as all of those models were biologically plausible. Akaike’s Information Criteria for small sample sizes (AICc) was used for model selection, considering the models with ∆AICc < 2 to be the most parsimonious and best supported [41]. A model average procedure was applied to ascertain the significance of the covariates included in the selected models [41,42]. The significance level for the coefficients was set at α = 0.05.

In order to identify the minimum distance that significantly increased recolonisation time, distances to the nearest colonies were grouped into three levels: A (<500 m), B (500–1000 m) and C (>1000 m). This was tested via the analysis of variance test (ANOVA), including the distance levels as a fixed factor and Ln (Rt) as a dependent variable. Finally, we conducted Tukey’s post hoc test to reveal differences between pairs of distance levels. The significance level for these analyses was set at α = 0.05.

All analyses conducted and figures constructed (except Figure 1) employed RStudio [43] via the R packages car, MuMIn and ggplot2 [44,45,46].

## 3. Results

The Rt was highly variable among buildings and ranged from 50 to 3072 days (mean ± SE: 651.1 ± 197.9 days). Two of the evaluated models revealed ∆AICc < 2, which accounted for a 92% cumulative weight (Table 1). The distance was included in both models, but Pigeon presence was only included in the second model (Table 1). Model averaging showed a significant positive effect of distance on Rt of Feral Pigeons (Table 2, Figure 2). On average, Rt was higher in buildings where no Feral Pigeons were observed at the end of the control programme (mean ± SE: 911.9 ± 347.8 days) than it was in buildings where Feral Pigeons were present (390.4 ± 165.7 days; Figure 3), although this was not statistically significant (Table 2). The Rt varied among distance levels (ANOVA: F = 8.28, 2 df, *p* = 0.005), with the post hoc Tukey test showing a significant difference between buildings closest to Pigeon source areas (<500 m) and buildings that were >500 m away from source areas. Thus, the buildings closest to source areas (<500 m) were recolonised more rapidly (mean ± SE: 136. 3 ± 49.3 days) than were the buildings that were >500 m away from source areas (500–1000 m: 721.7 ± 269.8; >1000 m: 1062.1 ± 381 days).

## 4. Discussion

Abundant research shows that the control of Feral Pigeon populations using methods based on the removal of individuals is ineffective at reducing abundance in the long term (see references cited in Stukenholtz et al. [36]). Most studies have indicated that several months, or even days [32], after the control campaign ended, there was a rebound effect and Feral Pigeon populations reached initial levels or, in certain cases, exceeded the initial abundance. The aforementioned studies were conducted primarily in the centres of large cities where Feral Pigeon populations are larger and reach the highest densities, giving them a high compensatory potential to minimise the effect of the extraction of individuals carried out during control programmes [32,36]. Conversely, the present study was conducted in residential areas where buildings were separated from each other by gardens, green areas, and golf courses. There was, therefore, a low density of buildings and numbers of Feral Pigeons were lower than in large cities [31]. In this setting, the removal of Feral Pigeons can be effective in the long term and can last for years, depending on the factors that modulate the recovery process. In our study, we focused our attention on two factors as the main modulators of the recolonisation process: the distance to a source area and the exhaustiveness of the control programmes. We should not, however, exclude the existence of other factors that could also affect recovery time, such as food availability in general and feeding by humans in particular [5]

The recovery of Feral Pigeon colonies that have been subjected to control measures occurs due to a high reproductive rate in the remaining colony (resulting from high levels of recruitment of young Pigeons), the early incorporation of young Pigeons into the breeding colony, an extended breeding season (which lasts for the entire year), and low predation rates [2,3,7,47]. Nevertheless, reproduction may be limited when a control programme minimises the number of survivors. In our study area, recovery time tended to be longer in buildings where no Pigeons were observed once the control programme concluded, although this difference was not statistically significant. This could be because the exhaustiveness of the control programmes was high in all the buildings, with the highest abundance of Feral Pigeons at the end of the control programme being only eight individuals. The immigration of individuals from uncontrolled areas may be the primary mechanism that contributes to the most rapid recovery cases [34], however, particularly in high-density colonies in which immigration rates are higher [48]. In our study area, the factor that most influenced the recovery time of Feral Pigeon colonies was the distance to other uncontrolled colonies that acted as source areas. Moreover, recovery time increased significantly with increasing distance to source colonies. Thus, the average recovery time of the colonies was approximately 4.5 months in buildings with source colonies at a distance of <500 m. In contrast, the average recovery time was approximately five (24 months) or eight (35 months) times longer if the distance to the source colony was greater than 500 m (i.e., 500–1000 category) or 1000 m, respectively.

This direct relationship between distance to source area and recovery or colonisation time has been demonstrated in other studies on bird communities in regions burned by wildfires. Furthermore, the results of the present study corroborate the results obtained in studies by Brotons et al. [49] and Watson et al. [50], which revealed that the recolonisation rate of birds in burned regions decreased as the distance to potential sources of colonisers increased.

Immigration appears to play a key role in the recovery of populations of other vertebrate pest species, such as the common vole (*Microtus arvalis*), that have been subjected to control programmes [51]. Accordingly, the recovery time of populations is not primarily driven by the reproduction rate of the surviving residents when there are source areas nearby. When source areas are more distant, however, reproduction may become the primary recovery mechanism, slowing down the recovery process. This may explain the results obtained in one study for buildings located >500 m from a source area, in which the immigration rate was extremely low or absent [34]. Furthermore, the low density of the colonies analysed in the present study is a potential factor contributing to an increase in the recovery time since immigration rates have been shown to decrease in low-density populations [48]. Additionally, reproduction may be limited when a control programme minimises the number of survivors. Although the results in the present study were not statistically significant, recovery time was longer in buildings where no Feral Pigeons were observed once the control programme concluded. Extending the duration of the control programme until the smallest possible population size is reached may therefore aid in increasing the recovery time, although this is not a determining factor.

A key factor contributing to the short-term effect of control measures in large cities is that these measures are generally developed at a local level (i.e., district and neighbourhood levels), and an entire city is not considered as a single management unit [32,34]. In most cases, the main objective of planned control campaigns in urban areas is to eliminate or reduce Feral Pigeon colonies in buildings where Pigeon extraction is carried out, while actions at the population level are practically non-existent. Under these circumstances, a local reduction in the density of the colony would be compensated for by incoming Pigeons that would rapidly recolonise the area [34]. It is therefore essential to identify effective management areas that increase the efficiency of control programmes. The results of our study suggest that in residential areas where colonies are more isolated, a minimum effective management unit is guaranteed when there are no uncontrolled Pigeon colonies within at least a 500 m radius of the building where the control measures are being conducted. These results are in accordance with the described patterns of daily movement and the use of space for urban Feral Pigeons. In European urban environments, Feral Pigeons generally show daily displacements, which rarely exceed 300–500 m [34,37]. Additionally, in a separate study, the exchange of Feral Pigeons between colonies, i.e., immigration, did not occur at a mean distance of 509 m (range: 306–775 m) [34]. The results of the present study demonstrated that at a distance of >500 m, the immigration rate is significantly reduced, which resulted in an increase in recovery time and a delay in the effect of the control measures over time. These findings highlight the relevance of identifying a management unit to implement control programmes and increase the long-term effectiveness of control measures. According to our results, in low-density urban areas, the management unit should include buildings with Feral Pigeon colonies that are less than 500 m apart.

## 5. Conclusions

We are aware of the limitations of our study, which are a direct consequence of the lack of a planned experimental design to address the aspects discussed. Because there are no previous such studies, however, our results provide a first step that will allow for an improvement in the efficiency of managing Feral Pigeon colonies in areas with a low density of buildings. In fact, our results suggest that in low-density urban areas, the recovery time of Feral Pigeon colonies is significantly determined by their distance from other colonies that can act as a source. We, therefore, recommend that when control programmes are conducted in various buildings with active Feral Pigeon colonies, a buffer of at least a 500 m radius should be maintained around these buildings. Thus, the probability that nearby colonies can act as a source is reduced, and, consequently, the recovery time will be increased. Although the relationship between Feral Pigeon abundance at the end of the control program and recovery time was not statistically significant in our study, recovery time was higher on average in buildings where no Feral Pigeons were observed at the end of the control programme. For this reason, we also recommend extending the duration of a control programme until the density of Feral Pigeons is reduced to the minimum possible, as this will increase the recovery time and delay the appearance of issues associated with increasing colonies of Feral Pigeons.

Further work is clearly needed to improve the effectiveness of control programmes in low-density urban areas. A first step could be to design experiments to test if control programmes carried out by management units are more effective than those carried out in isolated buildings without considering the proximity of source areas. Another step would be to confirm that the first Feral Pigeons to recolonise the controlled buildings come from the closest uncontrolled colonies. Finally, more research should be conducted to investigate the effect of Feral Pigeon abundance at the end of the control programme on recolonisation time.

## Figures and Tables

**Figure 1 animals-12-01056-f001:**
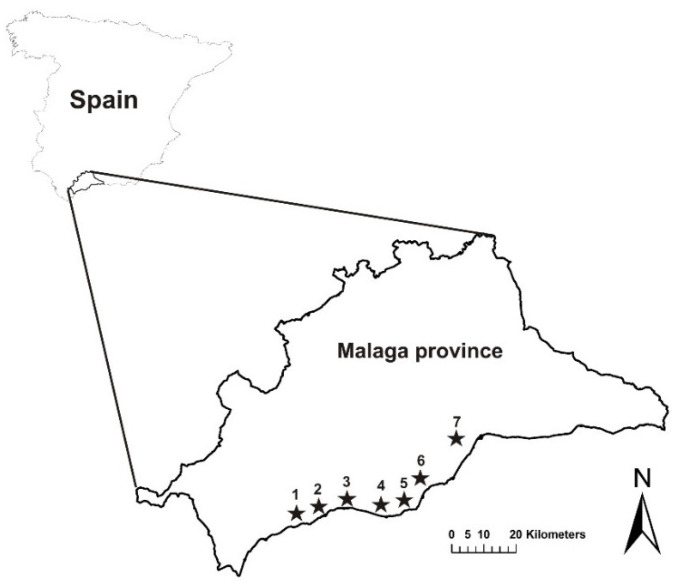
Study area and schematic location of the urban areas where a Feral Pigeon control programme was implemented (black stars). The numbers represent the location of each building in the study area (Map_ID; see Appendix A). This figure was prepared in ArcGIS 10.6 (Geographical Information System, ESRI, https://www.arcgis.com/ (accessed on 8 November 2021)).

**Figure 2 animals-12-01056-f002:**
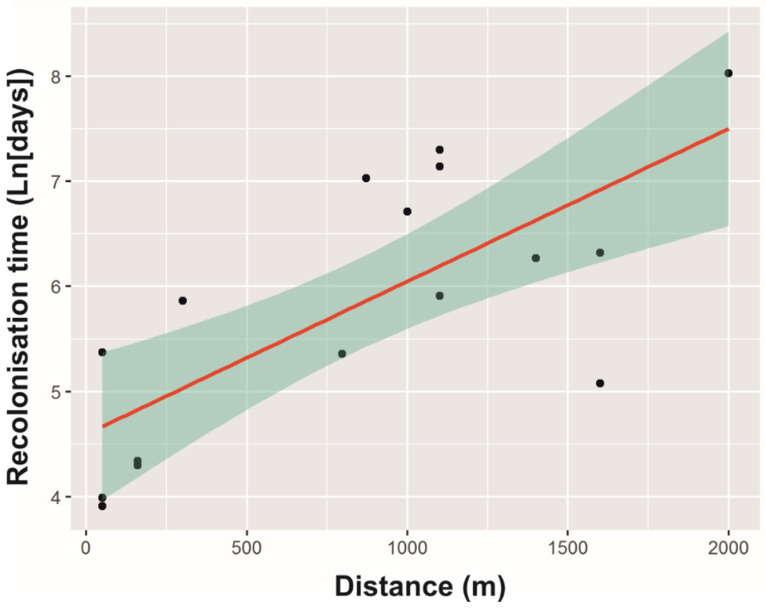
Relationship between the recolonisation time, expressed as Ln (days), and the distance in metres to the nearest colony of Feral Pigeons. A 95% credible interval is denoted by the shaded area.

**Figure 3 animals-12-01056-f003:**
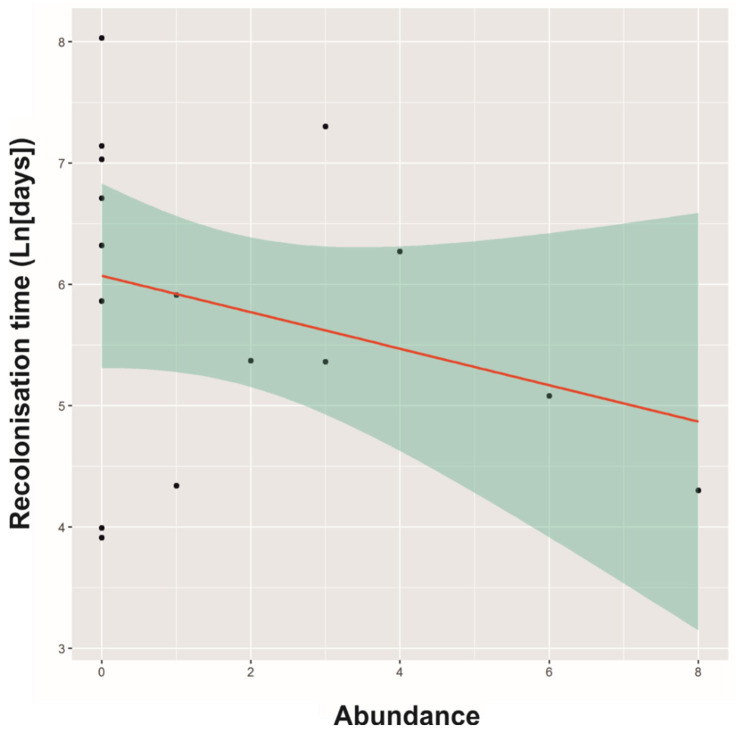
Relationship between the recolonisation time, expressed as Ln (days), and the abundance of Feral Pigeons remaining at the end of a control programme. A 95% credible interval is denoted by the shaded area.

**Table 1 animals-12-01056-t001:** The ranking of models investigating the recolonization time of Feral Pigeons based on AICc descriptions. Model parameters and estimates are presented for all possible models. The variables are Distance: the distance to the nearest colony of Feral Pigeons (m), Pigeons: the abundance of Feral Pigeons at the end of the control programme, and Distance * Pigeons: the interaction between the covariates.

Model	d.f.	logLik	AICc	ΔAICc	Weight
Distance	3	−19.737	47.5	0	0.426
Distance + Abundance	4	−18.014	47.7	0.2	0.387
Distance + Abundance + Distance * Abundance	5	−16.59	49.2	1.7	0.181
Null model	2	−25.944	56.8	9.3	0.004
Abundance	3	−25.227	58.5	11.0	0.002

**Table 2 animals-12-01056-t002:** The model-averaged coefficients and standard errors (SE) of the variables included in the best models that described the recolonisation time of the Feral Pigeons (i.e., AICc < 2). The asterisk (*) represents the interaction.

Variable	Estimate	SE	z	*p*-Value
Intercept	4.673	0.411	10.453	<0.0001
Distance	0.001	0.0003	3.689	<0.0003
Abundance	−0.524	0.438	1.086	0.277
Distance * Abundance	−0.00003	0.00009	0.37	0.71

## Data Availability

The data presented in this study are available on request from the corresponding author.

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
