# Peer review of "Source Areas as a Key Factor Contributing to the Recovery Time of Controlled Feral Pigeon (Columba livia var. domestica) Colonies in Low-Density Urban Locations"

_animals, 2022, doi:10.3390/ani12091056_

Round 1

Reviewer 1 Report

I put my comments directly on the manuscript. Well, pigeons are tracked as pests, is true, but it probably needs also few more words on ethics of the study.

And the second important point, I see potential limits of the manuscript and design of this (semi-) experiment but is well expressed by the authors

Author Response

I provided an example from Poland

In the new version of the manuscript we have rewritten the expression "no studies" for "very few studies" (lines 14 and 26).

Italized Latin names

Done (line 40).

It very much depends on spatial scale - pleace check paper - in references - Hetmanski et al, Przybylska et al

Now, we have rewritten the sentence to take into account the fact that predation is related with the areas of the cities and towns. We have added the two references highlighted by the reviewer (line 46).

Nice develop also here: Spennemann, D. H., & Watson, M. J. (2017). Dietary habits of urban pigeons (Columba livia) and implications of excreta pH–a review. European Journal of Ecology, 3(1), 27-41.

We agree with the reviewer. Therefore, in the new version of the manuscript we already included the reference Spennemann & Watson (2017) (lines 53 y 55).

Well, is probably true for some industrial areas located in suburban zones, but is only practical "action", effective, but without and scientific documentation, as I personally know from PL and CZ

We appreciate this comment with which we agree. Your observations for PL and CZ are similar to our case study. Although these control actions is a relative common practice, there are no scientific studies that have evaluated it. Our study highlights this fact and attempts to provide a first empirical approach.

Italized

Done (line 258).

Please consider, especially in term of methods:

Hetmański, T., Bocheński, M., Tryjanowski, P., & Skórka, P. (2011). The effect of habitat and number of inhabitants on the population sizes of feral pigeons around towns in northern Poland. European Journal of Wildlife Research, 57(3), 421-428.

Przybylska, K., Haidt, A., Myczko, Ł., Ekner-Grzyb, A., Rosin, Z. M., Kwieciński, Z., ... & Skórka, P. (2012). Local and landscape-level factors affecting the density and distribution of the Feral Pigeon Columba livia var. domestica in an urban environment. Acta Ornithologica, 47(1), 37-45.

Thank you for your comment. In the new version of the manuscript we have included both references in the methodology section (lines 143-144).

Reviewer 2 Report

This is an important subject and I think this paper is interesting and has potential, but in my view it requires major revision before publication. In general, I would recommend the authors reframe the paper in terms of a central question or hypothesis – such that they expect a correlation (in what terms) between the distance to a source population and the “recovery time” of a colony that has been removed. I think a hypothesis-driven paper would be much more powerful than the current approach, and if the authors choose to reframe the paper in this way, they should also provide some basis for their hypothesis (past research that led them to a particular prediction or expectation, or other logical basis). They do provide some interesting background in the introduction that would be useful towards this end, but revealing and additional details would enrich the paper substantially.

The authors do frame their inquiry in terms of a hypothesis in the methods section (lines 119-131) but I suggest using such a framework to revise the entire manuscript and clarifying aspects of their hypothesis that are currently quite vague. For example, they hypothesize that recovery/recolonization is modulated by two key factors: distance to the source and “the occurrence of pigeons” by which they mean remaining birds following the conclusion of the control program. Another way to express this might be as the completeness or exhaustiveness of the control program. Although it seems intuitive that the more birds remain following the end of control efforts, the more quickly they might recolonize/recover, the authors state there was no (statistical) difference between buildings where birds remained and where they did not – this seems surprising, yet the authors do not discuss this at any length. I suggest the authors get more clear and specific regarding their hypothesis as well; for example, they could introduce earlier the fact that pigeons typically travel less than 500 m (mentioned on line 253 of the discussion) and use this as a basis to predict that avoiding “source” areas within that distance would significantly extend the recolonization time. This is just one suggestion, but generally being more thoughtful, clear, and specific in introducing their predictions would make this a much stronger paper in my view.

In the simple summary/abstract, the authors write that their objectives were to evaluate the contribution of 1 ) total or partial removal of the previous existing colony on the “recovery time” of the feral pigeon colony (lines 16-17) and 2) distance to feral pigeon “source areas” (lines 18-19). Then, in the methods, the authors discuss the same factors in reverse order (117-131). Maintaining a consistent order throughout the manuscript would be helpful for clarity and comprehension so I recommend the authors do so. Reading the abstract, I wondered why the authors did not consider total or partial removal separately. Can you provide the reasons for this – is this because the outcome (whether the removal was total or partial) was not known until the surveys were done? Even if so, it stands to reason that a total removal would be more effective than a partial removal, I was surprised this was not addressed in the abstract.

I think the submitted manuscript has the makings of a good publication, and that revising the manuscript to reframe it as discussed above and providing a richer introduction and discussion would go a long way to improve it. There are also some issues with the English, for example in lines 20-22 and 35-36, in which they authors state their findings “highlight the relevance of identifying an effective management unit”… what exactly is this management unit, if you have identified one? Please explain. Also, in the same sentence, the authors refer to “control programs to… increase long-term effects.” I believe the authors mean instead to say “increase long-term effectiveness [of control programs].” Because many casual readers will only read the title and abstract/simple summary, the quality of the writing must be improved and the reader should be able to have a clear “take-home message” from this reading.

Finally, it does not seem at all surprising that the distance to the nearest “source area” would have an important effect on feral pigeons’ ability to recolonize a site that has been the subject of control efforts. While it is interesting that the authors have quantified this, I suggest the authors explore the data available to them further to see if anything else could be learned and included in the paper that would enhance its value to readers and management efforts. They state, and I agree, that no/few studies address this particular situation (line 264) and that underlines the novelty of their approach. Thus, the manuscript merits a more thorough treatment throughout, in my view, with more extensive background in the introduction, more careful presentation of the approach, and more thorough discussion of the findings. As it is, the discussion seems to repeat a few brief points from the introduction (lines 203-206 compared to lines 42-25). For the reader who has no background in this material, a adding some additional paragraphs to discuss these points in light of the paper’s objectives and findings would add to the value of the manuscript.

As mentioned, I suggest the paper be substantially revised, including extending the material, and reframing their approach, and ideally drawing out some additional conclusions that could be applied to management as well as suggests for future investigations/next steps. I also include here a few small questions/comments on particular lines:

Line 49 states that feral pigeons…”constitute a serious threat to human healt[h].” Can you provide some examples to support this? Are there known cases where the presence of pathogens carried by pigeons is known to have caused damage to human communities? Or are the main objections rather aesthetic/nuisance-related? It would be interesting to see a paragraph or two summarizing some research on this. As a specialist to some extent on problems related to invasive species, it has struck me that despite their abundance, feral pigeons do not generally have the reputation of being a threat to native birds, perhaps because they are generally so restricted to urban areas outside of their original native range (southern Europe, North Africa, etc). Because the native range of Rock Doves coincides with the feral pigeons in this study, is there any interaction between feral pigeons and Rock Doves that is detrimental to wild Rock Doves? Is there any detrimental effects on other native birds or wildlife? Examples of transmission of the above-mentioned pathogens to other birds or wildlife? These are just a few questions that could be added to provide support for management/control programs.

Lines 72-74 defines “recovery time” as the amount of time it takes for feral pigeon density to recover to a level where it “generates issues” – this statement should be revised to be more specific/detailed about what is meant by this and how one measures this.

Lines 217-222 discuss related research in “burned regions” – does this refer to controlled burns, wildfires, or something else? Please be more specific and clear and add some detail for those not familiar with this work.

I hope these comments are helpful and would be glad to review a revised version of the manuscript. Thank you for the opportunity to provide these comments and for this interesting project. 

Author Response

This is an important subject and I think this paper is interesting and has potential, but in my view it requires major revision before publication. In general, I would recommend the authors reframe the paper in terms of a central question or hypothesis – such that they expect a correlation (in what terms) between the distance to a source population and the “recovery time” of a colony that has been removed. I think a hypothesis-driven paper would be much more powerful than the current approach, and if the authors choose to reframe the paper in this way, they should also provide some basis for their hypothesis (past research that led them to a particular prediction or expectation, or other logical basis). They do provide some interesting background in the introduction that would be useful towards this end, but revealing and additional details would enrich the paper substantially.

We thought that our approach leads the reader to the main object of study that we wanted to highlight, which is the importance of investigating the recovery time of feral pigeon colonies in low-density urban areas where a control campaign was previously conducted. However, we think that highlighting a central question or hypothesis is a positive contribution that allows us to precisely define the aspects that we want to test. For this reason, following the reviewer’s recommendation, in the new version of the manuscript we have explicitly stated the hypothesis that we want to test (lines 84-90 and 132-150). Now the assumption that underlies our hypothesis based on the factors that we consider key to explain the recovery time of the feral pigeon populations is clear from the beginning of the manuscript.

The authors do frame their inquiry in terms of a hypothesis in the methods section (lines 119-131) but I suggest using such a framework to revise the entire manuscript and clarifying aspects of their hypothesis that are currently quite vague. For example, they hypothesize that recovery/recolonization is modulated by two key factors: distance to the source and “the occurrence of pigeons” by which they mean remaining birds following the conclusion of the control program. Another way to express this might be as the completeness or exhaustiveness of the control program. Although it seems intuitive that the more birds remain following the end of control efforts, the more quickly they might recolonize/recover, the authors state there was no (statistical) difference between buildings where birds remained and where they did not – this seems surprising, yet the authors do not discuss this at any length. I suggest the authors get more clear and specific regarding their hypothesis as well; for example, they could introduce earlier the fact that pigeons typically travel less than 500 m (mentioned on line 253 of the discussion) and use this as a basis to predict that avoiding “source” areas within that distance would significantly extend the recolonization time. This is just one suggestion, but generally being more thoughtful, clear, and specific in introducing their predictions would make this a much stronger paper in my view.

Following the reviewer's recommendation we have modified the expression "the occurrence of pigeons" by this other "The exhaustiveness of the control program" (lines 141-142). In addition, we have included a paragraph in the discussion section about the relationship between the recovery time and feral pigeon abundance at the end of the control program (lines 236-241). We now argue that the absence of significant differences between the recovery time and feral pigeon abundance can be explained by the high exhaustiveness of the control program in our study area.

In response to the reviewer's previous comment, we have already clarified that in the new version of the manuscript we have rewritten the introduction section to explicitly state the hypothesis to be tested and scientific bases for that (lines 84-90).

In the simple summary/abstract, the authors write that their objectives were to evaluate the contribution of 1 ) total or partial removal of the previous existing colony on the “recovery time” of the feral pigeon colony (lines 16-17) and 2) distance to feral pigeon “source areas” (lines 18-19). Then, in the methods, the authors discuss the same factors in reverse order (117-131). Maintaining a consistent order throughout the manuscript would be helpful for clarity and comprehension so I recommend the authors do so. Reading the abstract, I wondered why the authors did not consider total or partial removal separately. Can you provide the reasons for this – is this because the outcome (whether the removal was total or partial) was not known until the surveys were done? Even if so, it stands to reason that a total removal would be more effective than a partial removal, I was surprised this was not addressed in the abstract.

The reviewer is right. To maintain consistency throughout the manuscript, now we have presented the factors in the same order in the different sections of the manuscript (simple summary/abstract, introduction and material and methods).

On the other hand, we have analysed the effect that the total or partial elimination of the pigeons has on the recovery time. In fact, it is one of the aspects to test, because as the reviewer comment, we think that there is a positive relationship between the recovery time and the completeness of the control program since the lack of individuals prevents reproduction. Now we discuss these implications in lines 236-241).

I think the submitted manuscript has the makings of a good publication, and that revising the manuscript to reframe it as discussed above and providing a richer introduction and discussion would go a long way to improve it. There are also some issues with the English, for example in lines 20-22 and 35-36, in which they authors state their findings “highlight the relevance of identifying an effective management unit”… what exactly is this management unit, if you have identified one? Please explain. Also, in the same sentence, the authors refer to “control programs to… increase long-term effects.” I believe the authors mean instead to say “increase long-term effectiveness [of control programs].” Because many casual readers will only read the title and abstract/simple summary, the quality of the writing must be improved and the reader should be able to have a clear “take-home message” from this reading.

Thanks for the comment. Following the reviewer's recommendation, we have included a phrase identifying the management unit according to our results (lines 292-295). In addition, in the new conclusions section we remark: “we recommend that control programs should be conducted in various buildings with active pigeon colonies with a buffer of at least a 500 m radius from these buildings. Thus, the probability that nearby colonies can act as a source is reduced and, consequently, the recovery time will be increased” (lines 303-305).

We have replaced the expression "… control programs and increase the long-term effects of control measures" with "control programs and increase long-term effectiveness of control measures" (lines 293-294).

The texts of the new version of the manuscript have been revised by Cambridge Proofreading & Editing LLC.

Finally, it does not seem at all surprising that the distance to the nearest “source area” would have an important effect on feral pigeons’ ability to recolonize a site that has been the subject of control efforts. While it is interesting that the authors have quantified this, I suggest the authors explore the data available to them further to see if anything else could be learned and included in the paper that would enhance its value to readers and management efforts. They state, and I agree, that no/few studies address this particular situation (line 264) and that underlines the novelty of their approach. Thus, the manuscript merits a more thorough treatment throughout, in my view, with more extensive background in the introduction, more careful presentation of the approach, and more thorough discussion of the findings. As it is, the discussion seems to repeat a few brief points from the introduction (lines 203-206 compared to lines 42-25). For the reader who has no background in this material, a adding some additional paragraphs to discuss these points in light of the paper’s objectives and findings would add to the value of the manuscript.

Following the reviewer's recommendation, we have included in the introduction a sentence in which we highlight that no studies have been carried out that address the recovery time of the colonies after a control program or the minimum management area to increase the success of the control campaign (lines 74-77). In addition, we have modified the discussion to treat the findings of our study in a more extensive and detailed way (lines 235-241, 294-295).

As mentioned, I suggest the paper be substantially revised, including extending the material, and reframing their approach, and ideally drawing out some additional conclusions that could be applied to management as well as suggests for future investigations/next steps.

Following the recommendation of the reviewer we have included a new paragraph in the discussion section suggesting what could be the next steps to improve the effectiveness of control programs in low-density urban areas (lines 314-321).

I also include here a few small questions/comments on particular lines:

Line 49 states that feral pigeons…”constitute a serious threat to human healt[h].” Can you provide some examples to support this? Are there known cases where the presence of pathogens carried by pigeons is known to have caused damage to human communities? Or are the main objections rather aesthetic/nuisance-related? It would be interesting to see a paragraph or two summarizing some research on this.

Yes, overall the main objections are related to aesthetic and nuisance. As far as we know, there is no evidence of serious damage to the health of human communities. However, it has been shown that in certain circumstances people's health is compromised, especially when there is an overabundance of pigeons and the possibility of contact increases. We have included a sentence highlighting this, on basis to the reported references (lines 50-51). For example, Haag-Wackernagel (2006) states that in the city of Lucerne in the period 1941-2004, 207 transmissions of human pathogens from feral pigeons were reported. Thirteen of these reported illnesses proved fatal. On eleven occasions the affected persons were immunocompromised. In a previous study, Haag-Wackernagel and Moch (2004) also highlighted that immunocompromised people are the most susceptible to becoming ill due to the transmission of pathogens from pigeons.

As a specialist to some extent on problems related to invasive species, it has struck me that despite their abundance, feral pigeons do not generally have the reputation of being a threat to native birds, perhaps because they are generally so restricted to urban areas outside of their original native range (southern Europe, North Africa, etc). Because the native range of Rock Doves coincides with the feral pigeons in this study, is there any interaction between feral pigeons and Rock Doves that is detrimental to wild Rock Doves? Is there any detrimental effects on other native birds or wildlife? Examples of transmission of the above-mentioned pathogens to other birds or wildlife? These are just a few questions that could be added to provide support for management/control programs.

Graczyk et al. (2007) have shown that the faeces of feral pigeon contain high concentrations of Enterocytozoon bieneusi spores that affect different groups of vertebrates such as fish, birds and mammals.

We are not aware that the domestic pigeon has a detrimental effect on the Rock Dove. In southern Europe, the Rock Dove has a very localized distribution linked mainly to coastal rocks and does not usually coincide spatially with the feral pigeon.

Lines 72-74 defines “recovery time” as the amount of time it takes for feral pigeon density to recover to a level where it “generates issues” – this statement should be revised to be more specific/detailed about what is meant by this and how one measures this.

To avoid any confusion, in the new version of the manuscript we have rewritten the definition of recovery time so that the meaning is clear and concise (lines 74-76).

Lines 217-222 discuss related research in “burned regions” – does this refer to controlled burns, wildfires, or something else? Please be more specific and clear and add some detail for those not familiar with this work.

Done. We now specify that the burned areas are the result of wildfires (lines 252-253).

Reviewer 3 Report

This manuscript reports correlative results on the effects of the residual abundance of pigeons following individual removal for regulation and of the distance to the nearest first uncontrolled pigeon colony on the recovery time after individual removal (i.e. the time taken to reach the initial abundance of pigeon before pigeon removal for regulation). The problematic is particularly interesting because most of strategy used for animal regulation are not evaluated. Here, the manuscript addresses this interesting question on the efficiency of animal removal for regulation and on the factors affecting the efficiency of such a strategy. Unfortunately, the method used to evaluate abundances has not been described because it has been performed by a private wildlife management company. Therefore, it is difficult to trust into the data collected because there are not reproducible.

Furthermore, the correlative nature of the study cannot exclude alternative explanations of the relation found between the distance to the nearest colony and the recovery time. Authors should at least mention this point and discuss it in the discussion.

The method of data collection is not explicit. It is not clear in the main text that the replica involved the same building several times. As authors collected abundances on 11 buildings, you expected to have 11 dots on the figure 2 and 3 which is not the case (there are more, and the information can be found in the supplementary material). In statistical analyses, authors should therefore rerun the different models tested by involving the “building” effect nested into the site as a random factor and the site as random factor.

To improve statistical power and to be consistent with the figure 2, authors should include the distance to the nearest colony as a quantitative variable instead of a categorical factor (500<; 500-1000; >1000).

The figure 4 is redundant with the figure 3 and can be removed.

The discussion needs to be less affirmative as the study is correlative (especially for the residual abundance as the effect is not significant) and should focus on the immigration hypothesis and propose solution to test it. For instance, are we sure that pigeons participating for a shorter recovery time came from the nearest uncontrolled colony.

Author Response

This manuscript reports correlative results on the effects of the residual abundance of pigeons following individual removal for regulation and of the distance to the nearest first uncontrolled pigeon colony on the recovery time after individual removal (i.e. the time taken to reach the initial abundance of pigeon before pigeon removal for regulation). The problematic is particularly interesting because most of strategy used for animal regulation are not evaluated. Here, the manuscript addresses this interesting question on the efficiency of animal removal for regulation and on the factors affecting the efficiency of such a strategy. Unfortunately, the method used to evaluate abundances has not been described because it has been performed by a private wildlife management company. Therefore, it is difficult to trust into the data collected because there are not reproducible.

Regarding the method used to evaluate the abundance of pigeons we say the following: "At the end of the control program, field technicians recorded the abundance number of feral pigeons in the building where the control was conducted" (lines 129-131 in the original version, lines 144-146 in the new version). We have not described it in detail because this methodology has already been published in the study by Farfán et al. (2019).

Furthermore, the correlative nature of the study cannot exclude alternative explanations of the relation found between the distance to the nearest colony and the recovery time. Authors should at least mention this point and discuss it in the discussion.

You are right. We have included a new sentence in the discussion section where we discuss the existence of other factors that could affect the recovery time such as the supply of food by humans (lines 226-230).

The method of data collection is not explicit. It is not clear in the main text that the replica involved the same building several times. As authors collected abundances on 11 buildings, you expected to have 11 dots on the figure 2 and 3 which is not the case (there are more, and the information can be found in the supplementary material). In statistical analyses, authors should therefore rerun the different models tested by involving the “building” effect nested into the site as a random factor and the site as random factor.

Indeed, there are 16 points in total because there are four locations repeated over time. Therefore, we consider that each sampling, being carried out in different years, is an independent observation due to the inter-annual variability, so that we obtain 16 independent observations. All under a low building density context that all sites comply with. We have clarified this in the new version of the manuscript (lines 154-157).

We are aware that these analyses are simple, but as this is an observational study with a small sample size, we have no scope for more complex analyses. We agree with your suggestion to use the nested design in our study. Unfortunately our data set (n=16) prevent us to build complex models.

We are aware that our analyses are simple and do not include random effects that should be taken into account. However, we believe that our work has much room for improvement, but that the results shown can be the basis for future work in experimental designs to test our hypotheses and results with more powerful statistical analyses. In this sense, we have included in the discussion a paragraph highlighting this.

To improve statistical power and to be consistent with the figure 2, authors should include the distance to the nearest colony as a quantitative variable instead of a categorical factor (500<; 500-1000; >1000).

We have grouped the pigeons into distances (categorisation) to try to differentiate whether there is a minimum distance at which the pigeons recolonise before and on the basis of the average movements of the species in urban environments (less than 500 m). This allows us to establish a minimum distance at which management should be carried out, in what we have called the effective management area. With the linear approach this is not possible; we only obtain a positive relationship.

The figure 4 is redundant with the figure 3 and can be removed.

The reviewer may have been confused in comparing these two figures, which show different relationships with recolonization time: distance to source areas in intervals and abundances.

The discussion needs to be less affirmative as the study is correlative (especially for the residual abundance as the effect is not significant) and should focus on the immigration hypothesis and propose solution to test it. For instance, are we sure that pigeons participating for a shorter recovery time came from the nearest uncontrolled colony.

Much of the discussion focuses on the immigration hypothesis and its effect on the recolonization process. Now, in the new version, regarding the abundance, we say that the non-significant effect on the recovery time can be explained because the exhaustiveness of the control programs which was high in all the buildings (lines 235-241) and that more research should be done on the effect of pigeon abundance at the end of the control program on recolonization time (319-321).

Round 2

Reviewer 2 Report

I appreciate the authors' addressing my concerns and comments about the paper; I believe the paper merits publication after a final check by the editor. Thanks and congratulations!

Author Response

Thank you very much for your comments, which have considerably improved the first version of the manuscript.

In the new version of the manuscript the language and style have been reviewed by Cambridge Proofreading & Editing LLC.